# Prenatal Diagnosis of *PPP2R1A*-Related Neurodevelopmental Disorders Using Whole Exome Sequencing: Clinical Report and Review of Literature

**DOI:** 10.3390/genes14010126

**Published:** 2023-01-02

**Authors:** Tingying Lei, Li Zhen, Xin Yang, Min Pan, Fang Fu, Jin Han, Lushan Li, Dongzhi Li, Can Liao

**Affiliations:** Prenatal Diagnostic Center, Guangzhou Women and Children’s Medical Center, Guangzhou Medical University, Guangzhou 510623, China

**Keywords:** prenatal diagnosis, *PPP2R1A*, neurodevelopmental disorder (NDD), whole-exome sequencing (WES), macrocephaly, agenesis of corpus callosum (ACC)

## Abstract

*PPP2R1A*-related neurodevelopmental disorder (NDD) is expressed with autosomal dominant inheritance and is typically caused by a pathogenic de novo *PPP2R1A* mutation. It is characterized by the predominant features of hypotonia, developmental delay, moderate-to-severe intellectual disability, agenesis of corpus callosum (ACC), ventriculomegaly, and dysmorphic features; however, none of these anomalies have been diagnosed prenatally. We report on the prenatal diagnosis of *PPP2R1A*-related NDD in two fetuses by whole exome sequencing. Fetus 1 had partial ACC and severe lateral ventriculomegaly; the pathogenic heterozygous c.544C > T (p. Arg182Trp) de novo missense variant in *PPP2R1A* was detected. Fetus 2 had severe enlargement of the lateral and third ventricles and macrocephaly; they showed a heterozygous likely pathogenic mutation in *PPP2R1A* gene (c.547C > T, p. Arg183Trp). Both variants were de novo. This was the first study to use trio WES to prenatally analyze fetuses with *PPP2R1A* variants. Prenatal diagnosis will not only expand the fetal phenotype of this rare genetic condition but also allow for an appropriate counseling of prospective parents regarding pregnancy outcomes.

## 1. Introduction

Protein phosphatase 2A (PP2A) is a serine-threonine phosphatase in the human body that plays an essential regulatory role in a variety of cellular functions. It is a heterotrimeric protein composed of a scaffolding subunit (A), variable regulatory subunits (B), and a catalytic subunit (C) [1]. The *PPP2R1A* gene, located on chromosome 19q13.41, encodes the scaffolding Aα subunit, which is highly expressed in the developing and adult brain, with a moderate increase during brain development [2]. PPP2R1A is composed of 15 HEAT repeat motifs, of which HEATs 1–8 mediate interactions with a specific regulatory B subunit [3]. The reported causative variants in previous studies are located in HEAT domains 1, 4, 5, 6, and 7 of PPP2R1A and are associated with neurodevelopmental disorder (NDD) [4]. To date, fewer than 50 cases with pathogenic variants in the *PPP2R1A* gene have been described with typical features of severe, persistent hypotonia, developmental delay, agenesis of corpus callosum (ACC), ventriculomegaly, and dysmorphic features. However, none of them have been diagnosed prenatally. In our study, we report the prenatal diagnosis of *PPP2R1A*-related NDD in a fetus with partial ACC and severe lateral ventriculomegaly, and in another fetus with severe lateral and third ventriculomegaly and macrocephaly by trio whole exome sequencing (WES).

## 2. Materials and Methods

### 2.1. Case Presentation

The mother of Fetus 1 was a 32-year-old woman (gravida 1, para 0). The fetus had a low risk according to the results of the first trimester Down’s syndrome screening at 12 weeks of gestation, and the nuchal translucency (NT) measurement was 1.5 mm. At 22 + 3 weeks of gestation, the fetus’s lateral ventricles were enlarged during a second trimester ultrasound (left: 11.8 mm; right: 13.6 mm). However, the couple declined amniocentesis for genetic analysis. At 26 + 5 weeks of gestation, severe ventriculomegaly (left: 15.1 mm; right: 15.5 mm) was identified on an ultrasound scan (Figure 1A). No other intra/extra structural central nervous system (CNS) anomaly was discovered. Fetal magnetic resonance imaging (fMRI) was performed and found that the fetus had partial agenesis of corpus callosum (ACC) (Figure 1B). At 27 weeks of gestation, they received genetic counseling on the CNS anomalies and underwent cordocentesis in our center. The results of karyotyping and chromosome microarray analysis (CMA) were normal and all IgM serology for recent fetal infection was negative. The trio WES test was then carried out.

The mother of Fetus 2 was 28-year-old woman (gravida 1, para 0). At 12 weeks of gestation, a first trimester ultrasound scan showed that the fetus’ CRL was in line with the woman’s last menstrual period. The NT measurement was 2.75 mm, and the nasal bone was visible. The mother decided to undergo noninvasive prenatal testing (NIPT) for aneuploidy screening. Testing indicated low risk for trisomy 21, trisomy 18, and trisomy 13. The fetus’s lateral ventricles were enlarged during the second trimester as seen by ultrasound imaging at 20 + 1 weeks of gestation (left: 13 mm; right: 12 mm). A detailed ultrasound at 23 weeks of gestation revealed that the fetus had an intact corpus callosum but displayed ventriculomegaly (left: 15 mm; right: 14 mm) (Figure 2A). Additionally, the biparietal diameter (BPD, 66 mm) and head circumference (HC, 231 mm) measurements were >2 SD while the abdominal circumference (AC, 190 mm) and femur length (FL, 40 mm) were normal, indicating that the fetus had macrocephaly. The couple was nonconsanguineous and without any significant family history. Amniocentesis was performed. The results of karyotyping and CMA were normal. At 25 + 4 weeks of gestation, the ultrasound scan showed that the fetus had severe enlargement of the lateral and third ventricles (left: 16 mm, right: 14 mm, the third ventricle: 3.5 mm) and macrocephaly (BPD: 75 mm > 2SD, HC: 266 mm > 2SD). Fetal MRI was performed and confirmed severe ventriculomegaly without absence of the corpus callosum and any other intra-CNS anomalies (Figure 2B). The couple elected to pursue trios WES testing.

### 2.2. Whole Exome Sequencing

DNA samples of the fetus-parental trios underwent WES. Exonic sequences were enriched using Agilent SureSelect human exome capture probes (V6, Life Technologies, Waltham, MA, USA) according to the manufacturer’s protocol. The DNA library was sequenced on a HiSeq XTen or Illumina Novaseq 6000 system (Illumina, Inc., San Diego, CA, USA) to obtain 150-bp paired-end reads. Coverage for the samples was >99% at a 20× depth threshold. Additional details on the data analysis are provided in our previous study [5]. All the selected variants were then classified as (P) pathogenic, (LP) likely pathogenic, (B) benign, (LB) likely benign, or (VUS) variant of unknown significance according to the American College of Medical Genetics and Genomics (ACMG) guidelines [6]. Candidate variants were classified and reviewed by a multidisciplinary team, including clinical and molecular geneticists, bioinformaticians, genetic counsellors, imaging experts, neonatologists, and maternal fetal medicine physicians, to determine the relevance to the clinical phenotypes. All the clinically significant variants were confirmed by Sanger sequencing.

## 3. Results

The WES result of Fetus 1 revealed a heterozygous pathogenic *PPP2R1A* (NM_014225.5) missense variant (c.544C > T, p. Arg182Trp), which was not detected in either parent. In Fetus 2, a heterozygous likely pathogenic missense variant in *PPP2R1A* (NM_014225.5) was detected: c.547C > T (p. Arg183Trp), which was de novo. A diagnosis of *PPP2R1A*-related NDD was therefore made based on these findings. Both couples chose to terminate the pregnancy after careful consultation of what the structural abnormalities and diagnosis of NDD may mean for the development of the children. The autopsy was declined.

## 4. Discussion

*PPP2R1A*-related NDD, also known as autosomal dominant intellectual developmental disorder-36 (MRD36, MIM:616362), is expressed with autosomal dominant transmission pattern and typically caused by a de novo *PPP2R1A* pathogenic variant. The disorder was first described in 2015 by Houge et al. [7] in five unrelated children (three females; two males) ranging in age from 1 year to 5 years with strikingly similar phenotypes including hypotonia, developmental delay, severe intellectual disability, ventriculomegaly, agenesis or hypoplasia of the corpus callosum, and facial dysmorphism. In the study, they identified three de novo missense mutations (c.536C > T, c.544C > T, and c.773G > A) in the *PPP2R1A* gene using trio WES [7]. Since then, a total of around 50 MRD36 cases with 17 pathogenic variants in *PPP2R1A* have been identified, although none of them have been diagnosed prenatally [7,8,9,10,11,12,13]. Over the past 9 years, WES has become a more powerful clinical test for the prenatal diagnosis of fetuses with monogenetic disease, with a detection rate about 8.5–10.3% [14,15]. Thus, it was recommended to be a first-tier test used in fetuses with congenital abnormalities and normal results of karyotyping and CMA. Our findings are the first report of the prenatal diagnosis of MRD36 in two fetuses with partial ACC, severe lateral ventriculomegaly, and macrocephaly using trio WES. 

MRD36 is generally characterized by hypotonia, severe motor delay, mild to severe intellectual disability, severe speech delay, abnormal head size (macrocephaly or microcephaly), and brain malformations (complete or partial ACC, ventriculomegaly, periventricular leukomalacia) [7,10]. Craniosynostosis, seizures, ptosis, ear-shape abnormality, hearing loss, joint hypermobility, short stature, and scoliosis have also been reported [12]. However, almost all the reported *PPP2R1A*-related NDD patients were with an uneventful pregnancy. To date, only three cases were reported with abnormal structural findings in pregnancy. Wallace et al. [8] reported a MRD36 male infant with severe lateral and third ventriculomegaly in pregnancy. Ruxmohan et al. [11] present a 14-month-old MRD36 boy with congenital hydrocephalus. In the study by Melas et al. [12], they described a 16-month-old full-term MRD36 male with prenatally identified progressive microcephaly, ventriculomegaly, and global developmental delay. In our study, Fetus 1 displayed partial ACC and severe lateral ventriculomegaly, and Fetus 2 had severe lateral and third ventriculomegaly, and macrocephaly, by prenatal ultrasound/MRI examination. Therefore, our findings will not only expand the fetal phenotype of this rare genetic condition but also bring forward the diagnosis of the disorder and allow for the appropriate counseling of prospective parents regarding pregnancy outcomes.

As we all know, approximately 90% (35/39) of the patients with MRD36 has been reported with abnormal CNS imaging, including brain malformations and abnormal head size [4,11,12]. CNS abnormal findings according to recent studies are summarized in Table 1. Brain malformations, such as complete or partial ACC, ventriculomegaly, periventricular leukomalacia, delayed myelination, and pontocerebellar hypoplasia, are present in about 64.1% (25/39) of MRD36 individuals, with complete or partial ACC being the most common. In our study, Fetus 1 also displayed partial ACC. ACC is one of the most common brain malformations, with an incidence of approximately 1 in 4000 live births [16]. Follow-up studies showed that about 28.8% to 50% of children with ACC experienced general intellectual, executive, academic, social, and/or behavioral difficulties, and about 20% to 71.2% were functioning at a level comparable to typically developing children [17,18]. Genetic and environmental factors have been proposed to be involved in ACC; of the 30–45% of cases with ACC with an identified genetic cause, 20–35% are caused by a mutation affecting a single gene [19]. Nevertheless, few studies systematically addressing the presence of mutations using prenatal WES in fetuses with ACC are available. In de Wit’s study, they reported that the estimated detection rate of WES in fetuses with ACC with normal CMA results was between 50.0% and 13.3% [20]. In 2020, Heide et al. [21] was the first to use WES in 65 fetuses with ACC, and the detection rate of diagnostic genetic variants was 18%. In a previous study of our center, we also used WES in fetuses with ACC but without chromosomal anomalies, and the total detection rate was 34.0% [22]. In this study, the case with partial ACC is the first to be identified prenatally with a heterozygous pathogenic variant in *PPP2R1A* gene and further confirm the necessity of using WES in the fetus with ACC.

Abnormal head size, including microcephaly or macrocephaly, is reported in about 51.3% (20/39) of the cases with MRD36. Head size did not show a clear correlation with the presence of epilepsy or the degree of ID, but the macrocephalic individuals mostly had moderate ID [10]. In the study of Lennart et al. [10], they presented 30 patients with 16 different pathogenic variants in *PPP2R1A*, 11 of whom were with macrocephaly and encompassed variants in HEAT-repeats HR5 (p. Met180Thr/Val/Lys/Arg, p. Thr178Asn/Ser) and HR4 (p. Phe141Ile). They were the first to conclude that macrocephaly was only seen in individuals without B55α subunit-binding deficit, and these patients had less severe ID and no epilepsy. Conversely, more disruptive variants in HEAT-repeats HR7 (p. Arg258Ser/His), HR6 (p. Ser219Leu, p. Val220Met), and HR5 (p. Pro179Leu, p. Arg182Trp, p. Arg183Trp/Gln) with impaired B55α but increased striatin binding were associated with severe ID, epilepsy, ACC, and sometimes progressive microcephaly [4]. In our study, Fetus 2 was identified with a *PPP2R1A* missense variant (c.547C>T, p. Arg183Trp) and was characterized by macrocephaly and severe ventriculomegaly. However, according to the previous studies, the individuals with the *PPP2R1A* variant affecting p. Arg183Trp almost always had normocephaly, ventriculomegaly, severe ID, hypotonia, and epilepsy [8,10]. This might be due to the small sample size of the current studies, the termination of the pregnancy, which did not allow us to completely explore prenatal and postnatal clinical features, as well as the impact of incomplete penetrance, variable expressivity, and position effect. More efforts should be taken to confirm the prenatal and postnatal genotype-phenotype relationship of the *PPP2R1A* variants in the future. Additionally, these findings re-emphasized the challenges and importance of providing professional genetic counseling to patients.

In summary, we described the prenatal diagnosis of MRD36 in fetuses with partial ACC and severe lateral ventriculomegaly and macrocephaly. This was the first study to use trio WES to prenatally analyze fetuses with *PPP2R1A* variants. Prenatal diagnosis will not only expand the fetal phenotype of the *PPP2R1A*-related NDD, but also allows for the appropriate counseling of prospective parents regarding pregnancy outcomes.

## Figures and Tables

**Figure 1 genes-14-00126-f001:**
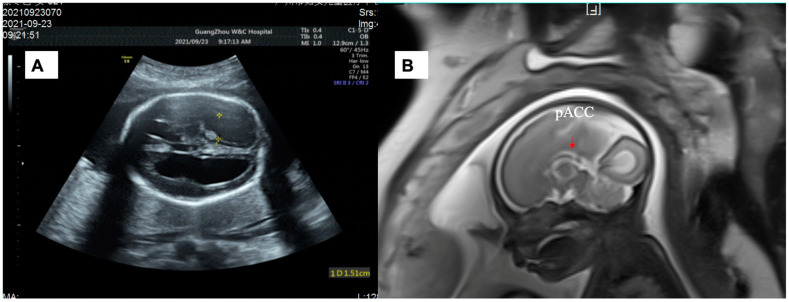
(**A**) Sonographic images of Fetus 1 showing lateral ventriculomegaly (left: 15.1 mm; right: 15.5 mm) at the 26-week scan. (**B**) Brain MRI images showing absence of the splenium of corpus callosum.

**Figure 2 genes-14-00126-f002:**
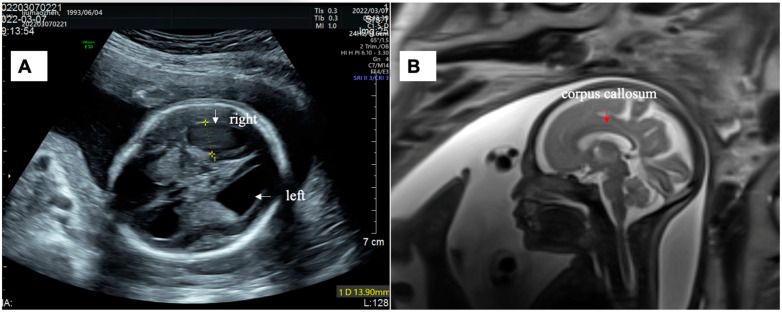
(**A**) Sonographic images of Fetus 2 showing lateral ventriculomegaly (left: 15 mm; right: 14 mm) at the 23-week scan. (**B**) Brain MRI images showing an intact corpus callosum.

**Table 1 genes-14-00126-t001:** Reports of abnormal CNS findings in patients with MRD36.

NO.	Age, Sex at Evaluation	DNA Change	Protein Change	Corpus Callosum	Head Size	Ventricle	Other CNS Findings	Epilepsy	Source
1	3.5 y, F	c.536C > T	p. P179L	Agenesis of the corpus callosum	-	-	-	-	[7]
2	4 y, F	c.544C > T	p. R182W	Hypoplasia of the corpus callosum	-	ventriculomegaly	-	yes	[7]
3	11 y, M	c.544C > T	p. R182W	Hypoplasia of the corpus callosum	-	-	-	yes	[7]
4	1 y, F	c.544C > T	p. R182W	Agenesis of the corpus callosum	-	ventriculomegaly	delayed myelination	yes	[7]
5	5 y, M	c.773G > A	p. R258H	Hypoplasia of the corpus callosum	-	-	delayed myelination	yes	[7]
6	Newborn, M	c.548G > A	p. R183Q	Hypoplasia of the corpus callosum	-	severe ventriculomegaly	pontocerebellar hypoplasia	yes	[8]
7	9 mo, M	c.656C > T	p. S219L	Agenesis of the corpus callosum	-	ventriculomegaly	reduced brain parenchyma, delayed myelination in the white matter, peripheral and central auditory impairment	yes	[9]
8	11 y, F	c.96C > G	p. I32M	-	-	-	-	yes	[10]
9	4 y, M	c.421T > A	p. F141I	-	macrocephaly	-	-	-	[10]
10	18 y, M	c.455C > T	p. S152F	-	-	-	-	-	[10]
11	18 y, F	c.532A > T	p. T178S	-	macrocephaly	-	-	-	[10]
12	12 y, M	c.533C > A	p, T178N	-	macrocephaly	-	-	-	[10]
13	4 y, M	c.536C > T	p. P179L	Agenesis of the corpus callosum	microcephaly	ventriculomegaly	-	yes	[10]
14	3 y 9 mo, M	c.539T > C	p. M180T	-	macrocephaly	-	-	-	[10]
15	6 y, F	c.539T > C	p. M180T	-	macrocephaly	-	-	-	[10]
16	2 y, F	c.539T > C	p. M180T	Hypoplasia of the corpus callosum	macrocephaly	-	delayed myelination	-	[10]
17	23 y, F	c.539T > C	p. M180T	-	macrocephaly	-	periventricular leukomalacia	-	[10]
18	2 y, M	c.539T > C	p. M180T	-	macrocephaly	-	-	-	[10]
19	27 y, M	c.538A > G	p. M180V	-	macrocephaly	-	-	-	[10]
20	20 y, M	c.538A > G	p. M180V	-	macrocephaly	-	-	-	[10]
21	1 y 4mo, M	c.538A > G	p. M180V	-	macrocephaly	-	-	-	[10]
22	10 mo, F	c.539T > A	p. M180K	-	-	-	-	yes	[10]
23	9 y, F	c.539T > G	p. M180R	-	microcephaly	-	-	-	[10]
24	6 y, F	c.544C > T	p. R182W	Agenesis of the corpus callosum	-	ventriculomegaly	-	yes	[10]
25	4 y, M	c.544C > T	p. R182W	Hypoplasia of the corpus callosum	-	ventriculomegaly, hydrocephalus	-	-	[10]
26	3 y, F	c.544C > T	p. R182W	Agenesis of the corpus callosum	-	ventriculomegaly	delayed myelination	yes	[10]
27	2 y, M	c.547C > T	p. R183W	-	-	ventriculomegaly, hydrocephalus	-	yes	[10]
28	20 y, M	c.656C > T	p. S219L	Hypoplasia of the corpus callosum	microcephaly	-	-	yes	[10]
29	2 y 4 mo, M	c.656C > T	p. S219L	Hypoplasia of the corpus callosum	-	ventriculomegaly	-	yes	[10]
30	7 y, M	c.656C > T	p. S219L		-	-	-		[10]
31	4 y, F	c.658G > A	p. V220M	Agenesis of the corpus callosum	-	ventriculomegaly	-	yes	[10]
32	7 y, F	c.658G > A	p. V220M	Agenesis of the corpus callosum	microcephaly	ventriculomegaly	-	yes	[10]
33	3 y, M	c.658G > A	p. V220M	Hypoplasia of the corpus callosum	-	-	periventricular leukomalacia, delayed myelination	-	[10]
34	4 y, M	c.658G > A	p. V220M	Hypoplasia of the corpus callosum		-	periventricular leukomalacia	-	[10]
35	4 y, M	c.773G > A	p. R258H	Hypoplasia of the corpus callosum	microcephaly	ventriculomegaly	-	yes	[10]
36	1 y 6 mo, M	c.773G > A	p. R258H	Hypoplasia of the corpus callosum	microcephaly	-	-	-	[10]
37	1 y 1 mo, M	c.772C > A	p. R258S	Hypoplasia of the corpus callosum	microcephaly	-	-	-	[10]
38	14 mo, M	-	-	hypoplastic/absent corpus callosum	macrocephaly	hydrocephalus	pontocerebellar hypoplasia	yes	[11]
39	16 mo, M	c.773G > A	p. R258H	-	microcephaly	ventriculomegaly and enlarged third ventricle	-	-	[12]
40	26+ gestation weeks, M	c.544C > T	p. R182W	partial agenesis of the corpus callosum	-	severe lateral ventriculomegaly	-	-	our study
41	25+ gestation weeks, M	c.547C > T	p. R183W	-	macrocephaly	severe lateral and third ventriculomegaly	-	-	our study

## Data Availability

All data are available upon reasonable request from the corresponding author.

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
