# Peer review of "Prenatal Diagnosis of PPP2R1A-Related Neurodevelopmental Disorders Using Whole Exome Sequencing: Clinical Report and Review of Literature"

_genes, 2023, doi:10.3390/genes14010126_

Round 1
Reviewer 1 Report
The manuscript of Lei et al. presents PPP2R1A-related neurodevelopmental 2 disorders in two probands diagnosed based on the WES prenatally. Parents decided to terminate pregnancy in both cases, after genetic counseling. These are the first two cases of PPP2R1A-related neurodevelopmental 2 disorders diagnosed prenatally.
It is a concise case report, presenting clearly both phenotype as well as genetic results. The introduction provides sufficient information. The results are sound and presented clearly. The clinical description is detailed.
Minor comments:
Please check if the references are in the right format.
Table 1 is useful but quite long. Maybe it would be better to include it in the supplement.
Please fill in the author contribution, funding etc.
Reviewer 2 Report
The authors describe two prenatal cases where a PPP2R1A related neurodevelopment disorder (mental retardation dominant 36) was identified by trio whole exome sequencing.
The report is concise, and clear. The authors provide a useful table (table 1) listing previously reported cases with abnormal central nervous system findings.
The report is interesting in that it shows the power of exome sequencing in diagnosing conditions such as hypoplasia/agenesis of the corpus callosum that can be caused by mutations in tens of genes. The prenatal presentation in their two PPP2R1A cases is in line with what one would expect, given what is already known about the disorder in postnatal cases.
Author Response
English language and style have been revised.
Reviewer 3 Report
English language and syntax require marked improvements throughout the whole text. I will provide suggestions and corrections for the present review, as I understand the difficulties encountered by non-native speakers, but will not tolerate poor English language after a major revision.
As an additional note that applies to the whole text, gene symbols (e.g PPP2R1A) and latin expressions (e.g. de novo) should be in italics. This should be edited through the manuscript.
Page1
Abstract
Line 8
“expressed in an autosomal dominant manner”
Replace the whole expression with “with autosomal dominant inheritance” or “with autosomal dominant transmission pattern”. Please edit similar expressions throughout the text in the same fashion.
Lines 9-11
“ caused by a pathogenic de novo PPP2R1 mutation; characterized with predominant features of hypotonia; developmental delay….features”
There is an abuse of semicolon (“;”) punctuation, which should be removed and replaced with commas (“,”) if not specified otherwise in the present review. Also, “by” is the correct preposition for the verb “characterized”. Please revise English grammar and syntax.
Line 12
“none of them”
Replace with “none of these anomalies”
Line 13-15
“the result of Fetus 1 who was … missense variant (c.544C>T)”
Replace with “Fetus 1 had partial ACC and severe lateral ventriculomegaly, and was detected with the pathogenic c.544C>T (NOTE: add protein change) de novo missense variant in PPP2R1A in heterozygosity.
Line 14
“lateral and third ventriculomegaly”
This expression should be reworded, here and elsewhere in the text, as it is used multiple times. “Enlargement of the lateral and third ventricles” might be a suitable substitute.
Line 18
“Prenatal diagnosis not only expanse the prenatal phenotypes”
This sentence has no meaning in English. I suspect ”expanse” should read “expands” or “expanded”, but even in that case it shows little-to-no connection to the subsequent sentence, so it should be thoroughly edited.
Line 19
“parents”
This is not an appropriate nomenclature when referring to an ongoing pregnancy. “Prospective parents” is the correct term.
Introduction
Lines 28-29
“, which is highly expressed in the developing and adult brain, and moderately increases during brain development2.”
Replace with “… in the developing and adult brain, with moderate increase during brain development [present ref]”
Page2
Case presentation
Line 46
“gravid 1, parity 0”
This should be replaced with “gravida 1, para 0”
Line 49
“bilateral ventricles”
Replace with “lateral ventricles”. This should also be emended elsewhere in the text (e.g. line 64)
Lines 51-52
“The ultrasound revealed that the fetus had severe ventriculomegaly(…) at 26+5 weeks of gestation”
The sentence could use better wording. I suggest “At 26+5 weeks of gestation, severe ventriculomegaly (left…right…) was identified at ultrasound scan”.
Line 59
“women”
Replace with the singular “woman”
Lines 66-67
“intact corpus callosum, and ventriculomegaly”
Replace “and” with “but displayed”
Lines 68-69
“indicating that the fetus had macrocephaly”
The authors should provide also information on the biometry of other fetal districts to compare head size with body size (abdomen, thorax) and limb bones lengths
Lines 70-71
“The results of karyotyping and CMA were unremarkable”
The sentence could use better wording.
Line 73
“fMRI”
It is not elegant to start a sentence with an acronym beginning with a lower-case letter. Please revise
Page 3
Figure 1, Panel B
Please select and enlarge the portion of the image showing the fetal brain, and rotate the image by 180°. The panel should only depict the sagittal section of the fetal brain, rotated.
Figure 2, panel B
For a better comparison, can you provide a T2-weighed fetal MRI image also for Fetus 2?
Line 72
“were subjected to”
Replace with “underwent”
Page 4
Lines 87-88
“What’s more, all the qualifying variants”
“what’s more” should be removed. “Qualifying variants” should be replaced with “candidate variants”
Results
“p.R182W”
In accordance with HGVS recommendation, the three-letter symbol for amino acids should be preferred to the single letter, when possible, even if it is not strictly necessary.
Line 100
“Unfortunately”
The adverb is ethically inappropriate and should be removed
Discussion
Line 203
“also term”
Incorrect wording. Can be replaced with “also known as”.
Lines 109-110
“There were… Trio WES”
Unclear meaning, please revise
Lines 110-112
“Since then… identified”
What were the sources you retrieved these data from? Literature only? Databases? Both sources should be considered and cited
Line 119
“Characterized with”
Replace with “characterized by”
Line 124
“PPP2RA1 – related patients”
Add “NDD” between “related” and “patients”
Line 126
“Baby boy”
Replace with “male infant”
Line 130
“was characterized with”
Replace with “displayed”
Lines 132-133
There are differences in text font and size. Please edit
Page 5
Line 134
“help the couples to decide whether to continue a pregnancy”
This statement does not reflect the aims of prenatal diagnosis and has serious ethical criticalities. Also, the decision for termination of pregnancy has different laws across different countries.
Line 136
“have”
Replace with “has”
Line 141
“while complete or partial ACC is the most common”
Replace with “complete or partial ACC being the most common.
Line 142
“Fetus 1 was also characterized with”
Replace with “Fetus 1 also displayed”
Lines 147-148
The data should be presented in better English. Please revise
Lines 149
“few studies addressing”
Replace with “few studies systematically addressing”
Lines 150-155
The presentation of systematic studies on the application of exome sequencing in corpus callosum anomalies is confusing, at best. Please revise
Table 1
Please edit the table so that the DNA change column displays variants on a single line. Also, cite the references by Arabic numbers, as in the text, rather than by author name
Page 7
Lines 162-163
“macrocephalic individuals mostly had moderate ID”
Add reference at the end of the sentence.
Line 166
“They were first to concluded”
Replace with “they were the first to conclude”
Line 177
“characterize with”
Replace with “characterized by”
Line 174
“almost showed normpocephaly”
Do the authors mean “had mild microcephaly” or do they mean “almost always had normocephaly”? Please edit
Lines 184-186
The sentences have obscure meaning, and there are inconscistencies in text font and size. Please revise extensively.
Lines 187-192
The sections (author contributions, funding, institutional review board, informed consnt, data availability) are blank. Please fill the required fields
